# Novel Compound Heterozygous Variations in *MPDZ* Gene Caused Isolated Bilateral Macular Coloboma in a Chinese Family

**DOI:** 10.3390/cells11223602

**Published:** 2022-11-14

**Authors:** Shuang Zhang, Fangxia Zhang, Juan Wang, Shangying Yang, Yinghua Ren, Xue Rui, Xiaobo Xia, Xunlun Sheng

**Affiliations:** 1People’s Hospital of Ningxia Hui Autonomous Region (Ningxia Medical University Affiliated People’s Hospital of Autonomous Region), Ningxia Eye Hospital, Ningxia Clinical Research Center on Diseases of Blindness in Eye, Yinchuan 750001, China; 2Eye Center of Xiangya Hospital, Hunan Key Laboratory of Ophthalmology, National Clinical Research Center for Geriatric Disorders of Xiangya Hospital Central South University, Changsha 410008, China; 3Department of Ophthalmology, Qingdao West Coast New District Central Hospital, Qingdao 266071, China; 4Gansu Aier Ophthalmology & Optometry Hospital, Lanzhou 730050, China

**Keywords:** macular coloboma, MPDZ, compound heterozygous mutation, ellipsoid zone, retina

## Abstract

Macular coloboma (MC) is a rare congenital retinochoroidal defect characterized by lesions of different sizes in the macular region. The pathological mechanism underlying congenital MC is unknown. Novel compound heterozygous variations, c.4301delA (p.Asp1434fs*3) and c.5255C>G (p.Ser1752Ter), in the multiple PDZ domain (MPDZ) proteins were identified via whole-exome analysis on the proband with isolated bilateral macular coloboma in a Chinese family. Segregation analysis revealed that each of the unaffected parents was heterozygous for one of the two variants. The results of the in silico and bioinformatics analysis were aligned with the experimental data. The knockdown of MPDZ in zebrafish caused a decrease in the ellipsoid zone, a destruction of the outer limiting membrane, and the subsequent RPE degeneration. Overall, the loss of MPDZ in zebrafish contributed to retinal development failure. These results indicate that MPDZ plays an essential role in the occurrence and maintenance of the macula, and the novel compound heterozygous variations were responsible for an autosomal recessive macular deficiency in this Chinese family.

## 1. Introduction

Macular coloboma (MC; OMIM: %120300) is a rare congenital retinochoroidal defect that occurs in the macular region, with a prevalence of 0.5–0.7/10,000 live births, and can lead to severe decreases in visual acuity [1]. It presents as an absence of the retina, choroid, and sclera with an elliptical or rounded atrophic pigmented scar in the macula of the retina [2]. The biological mechanism underlying congenital MC, which can be ophthalmoscopically observed as well-circumscribed, punched-out atrophic lesions of about 3–6 disc diameters (DDs) in the central part of the fundus, is unknown. It is believed to result from the failure of normal optic fissure closure during weeks 5 to 7 of development and may result from an incomplete differentiation of congenital infections or embryonic development abnormities [1,3].

The genetic changes caused by the pathogenesis of MC have not been studied well due to the low number of cases. Genetic studies have been carried out to investigate the causes of MC and simultaneous eye disease or systemic disease [4,5,6]. It was reported that *BEST1*, *RIMS1*, and *CLDN19*, identified as hereditary retinal dysplasia and degeneration (*rdd*) phenotype-related genes, are related to MC [7]. The chromosomal gene *MPDZ* (multiple PDZ domain crumbs cell polarity complex component, OMIM: 603785) is another gene previously deemed to be responsible for pathology in *rdd* chicks, and heterozygous variations in *MPDZ* were identified in human retinitis pigmentosa (RP) and Leber congenital amaurosis (LCA) [8]. In addition to *rdd*-related diseases, homozygous truncation and compound heterozygous nonsense variations in *MPDZ* have been mostly reported in multiple patients with congenital hydrocephalus [9,10,11]. As MPDZ is localized in the cell–cell adhesion site and is involved in the regulation of permeability, it is assumed that the variation of this protein can destroy such localization. A previous study also revealed that the loss of MPDZ impaired ependymal cell integrity and led to perinatal-onset hydrocephalus in mice [12]. Therefore, changes in cerebral spinal fluid (CSF) secretion can lead to hydrocephalus [8].

In this study, we confirmed the role of MPDZ in human MC through the identification of complex heterozygote variations in a Chinese MC family. To the best of our knowledge, no reports related to MPDZ showing an association with the autosomal recessive MC phenotype are available. Functional analysis revealed that the compound heterozygous loss-of-function variations could lead to developmental defects in the retina.

## 2. Materials and Methods

### 2.1. Patients

In line with the tenets of the Helsinki Declaration, our research was approved and examined by the Ethics Committee of the People’s Hospital of Ningxia Hui Autonomous Region (LSKY2016018). Prior to admission, all the participants provided written informed consent.

Two patients (Figure 1A; II:3 and II:1) and three unaffected family members (Figure 1A; I:1, I:2, and II:2) were recruited from the family. Detailed medical records were obtained from each participant. Ophthalmological examinations were carried out, including total ophthalmological examinations, best-corrected visual acuity (BCVA) tests, slit-lamp inspections, fundus evaluations, and optical coherence tomography (HD-OCT4000, Carl Zeiss Meditec, Dublin, CA, USA). Peripheral blood samples (5 mL) were collected from all the participants. Genomic DNA was extracted using a DNA blood kit (QIAmp; Qiagen, Valencia, CA, USA).

### 2.2. Whole-Exome Sequencing (WES)

WES was conducted as described previously [13]. The classification and analysis of novel and rare variants (MAF < 0.5%) for virulence were targeted according to the American College of Medical Genetics and Genomics (ACMG) guidelines [14]. That is, the genome libraries were constructed using the xGen Exome Research Panel v1.0 (Integrated DNA Technologies, Coralville, IA, USA) and sequenced on the HiSeq 4000 platform (Illumina, San Diego, CA) after checking the DNA purity and concentration. The quality of the raw data was checked using FASTQC (https://www.bioinformatics.babraham.ac.uk/projects/fastqc/, accessed on 20 August 2022). After filtering out low-quality reads and the adaptor reads, the remaining reads were mapped to the reference genome (GRCh37/hg19) using the Burrows–Wheeler aligner (BWA). We performed insertion and deletion realignment, quality recalibration, and variant calling using GATK. In addition, ANNOVAR was used for the annotation of the detected variants. Those with minor allele frequencies (MAFs) of >0.5% were filtered using databases, including dbSNP, the 1K Genome Project (1KGP, http://www.internationalgenome.org/data, accessed on 20 August 2022), the Exome Aggregation Consortium Browser (ExAC, http://exac.broadinstitute.org, accessed on 20 August 2022), and the Genome Aggregation Database (gnomAD, https://gnomad.broadinstitute.org/, accessed on 20 August 2022).

### 2.3. In Silico Analysis

In silico analysis was conducted as described in our previous report [15]. The prediction of the effect of a novel variation site of the MC-associated pathogenic genes on protein function was performed on four publicly available servers with bioinformatic prediction tools: Polyphen2 (http://genetics.bwh.harvard.edu/pph2, accessed on 20 August 2022), SIFT (http://sift.jcvi.org, accessed on 20 August 2022), PROVEAN (http://provean.jcvi.org/index.php, accessed on 20 August 2022), and MutationTaster (http://www.mutationtaster.org, accessed on 20 August 2022). When all the predictions were tested for pathogenicity, the variants were classified as probably pathogenic in combination with other evidence. Frameshift variants, nonsense variants, and variants with experimental evidence of loss of protein function were classified as pathogenic.

### 2.4. Sanger Sequencing

Eight noncoding SNVs upstream of the *PRDM13* gene have been reported to be involved in North Carolina macular dystrophy (NCMD), which is similar to MC in phenotype [16,17,18,19]. Hence, a Sanger sequence was taken to rule out the other noncoding genetic causes. These variants are chr6:100,040,906G>T; chr6:100,040,906G>C; chr6:100,040,974A>C; chr6:100,040,987G>C; chr6:100,041,040C>T; chr6:100,046,783A>C; chr6:100,046,804T>C; and chr6:100,046,940A>G. Two pairs of primers were designed (Fw: 5′-GAGAAGACTAGATCAGGCTTCTTC-3′, Rv:5′-CTCTCATTCTCTGATTTTTAC-3′ and Fw: 5′-TCCTATGTCTTGAGTGCTCTGC-3′, Rv:5′-AAATGCCCAACACATAACAGG-3′). The cycling conditions were as follows: 95 °C for 3 min, 95 °C for 30 s, annealing at 55 °C for 30 s, and an initial extension at 72 °C for 30 s; this was followed by 72 °C for 5 min, for a total of 35 cycles.

### 2.5. Whole-Exome Sequencing (WES) and Bioinformatics Analysis of Zebrafish Line

In this study, we used AB-line wildtype zebrafish. The larvae were maintained in E3 medium. All the zebrafish experiments were conducted according to the Animal Care and Use Committee-approved protocol of Ningxia Medical University. All the animal experiments were approved by the local ethics committee.

### 2.6. Whole-Mount In Situ Hybridization

For whole-mount in situ hybridization, the zebrafish embryos were treated with 0.2 mM phenylthiourea (Sigma) to inhibit melanization at 24 h post-fertilization (hpf). A digoxigenin-labeled (Roche, 11277073910) zebrafish MPDZ anti-sense RNA probe was transcribed in vitro from a DNA template using T3 RNA transcription polymerase (Roche, 11031163001). The embryos were fixed in 4% paraformaldehyde (PFA) at 4 °C, following a standard protocol described elsewhere [20]. Table 1 presents the sequences used.

### 2.7. Immunofluorescence Staining

The zebrafish embryos were fixed with 4% paraformaldehyde at 4 °C overnight and then embedded in paraffin. Next, 5 μm sections were cut from paraffin-embedded zebrafish embryos. After deparaffinization and rehydration in an ethanol series, the sections were blocked with 10% goat serum in PBS for 30 min at room temperature (RT). Next, they were incubated with primary antibodies against MPDZ (Affinity, DF13680, 1:50 dilution, Ancaster, ON, Canada) and ZO-1 (ThermoFisher, 33-9100, 1:100 dilution, Waltham, MA, USA) at 4 °C. On day 2, the sections were washed with PBS five times; they were cultured with goat anti-rabbit IgG (H + L) highly cross-adsorbed secondary antibody, Alexa Fluor 488 (ThermoFisher, A-11034, 1:200), and goat anti-mouse IgG (H + L) highly cross-adsorbed secondary antibody, Alexa Fluor Plus 555 (ThermoFisher, A32727, 1:200), in the dark at RT for 2 h and stained with aqueous DAPI (Abcam, ab104139, Carlsbad, CA, USA) for confocal imaging.

### 2.8. Generation of MPDZ Crispants

The *MPDZ* crispants were generated according to a rapid CRISPR/Cas9-based gene-knockout approach as described in [21]. The template of inducible RNAs targeting the *MPDZ* and scrambled regulatory genes were produced by annealing and elongation and transcribed in vitro using a T7 High-Yield RNA Transcription Kit (Vazyme Biotech; TR101-01, Beijing, China). The sgRNAs (final 1 µg/µL for each sgRNA) and Cas9 protein (final 1 µg/µL) (Novoprotein, E365- 01A, Summit, NJ, USA) were mixed in Cas9 buffer solution and incubated at 37 °C for 5 min to form a ribonucleoprotein complex (RNP). The microinjection was carried out by injecting 0.5–1 nL of the mixture into yolk cells in one-cell-stage embryos. Table 1 presents the oligonucleotides from Genscript.

### 2.9. Capped Message RNA In Vitro Transcription

Capped message RNA (mRNA) of human normal wildtype and mutant *MPDZ* was transcribed in vitro from wildtype and mutant *MPDZ* expression vectors (synthesized and cloned into pcDNA3.1-HA by Genscript, Piscataway, NJ, USA). The expression vectors were linearized by *Xho*I digestion, synthesized using the HiScribeTM T7 ARCA mRNA kit (with tailing) (NEB, E2060S), and purified using the RNA Clean & Concentrator-25 kit (ZYMO Research, D4003, Irvine, CA, USA). Briefly, 30 ng/µL of each mRNA was co-injected with the RNP mixture into one-cell-stage embryos in the yolk. For the injections, the embryos that died within 24 h of injection were excluded because such death probably resulted from unspecific causes.

### 2.10. Zebrafish Histology

The zebrafish embryos were fixed with 4% paraformaldehyde at 4 °C overnight. Then, 5 μm sections were cut from paraffin-embedded tissues. After deparaffinization and rehydration, H&E staining and stereomicroscopy were applied for histopathological analysis.

### 2.11. Cell Culture and Transfection

Retinal pigment epithelium (RPE) cell lines were used in this study and cultured as described in [22]. For the transfection of the plasmids or siRNA, jetPRIME in vitro DNA and an siRNA transfection reagent (PolyPlus Transfection, pt-114-15) were used following the manufacturer’s instructions. The siRNAs were ordered from RiboBio Company (Guangzhou, China) and are listed in Table 1.

### 2.12. Immunoblot Analysis and Immunofluorescence (IF) Staining

For immunoblotting analysis, the cells were harvested at 48 h after transfection. After two washes with cold PBS on ice, RIPA lysis buffer (MCE, HY-K1001) supplemented with protease inhibitors (MCE, HY-K0011) was added. Equal amounts of protein samples were loaded and separated using a 4–20% Bis–Tris Gel (Future, F11420Gel) and then transferred onto polyvinylidene fluoride (PVDF) membranes (Millipore-Merck, Burlington, MA, USA). The PVDF membranes were blocked with 5% nonfat milk in PBST for 1 h at RT and then incubated with indicated primary antibodies overnight at 4 °C with rotation. After several washes with PBST, HRP-conjugated secondary antibodies were added for 2 h at RT. The NCM ECL Ultra kit was used for signal detection. For IF staining, the RPEs were plated in glass-bottom cell-culture dishes; after 48 h transfections, the cells were washed with PBS twice, fixed in 4% PFA for 20 min, and permeabilized with PBS containing 0.5% Triton X-100 for 10 min at RT. For rhodamine phalloidin staining, a 1:200 dilution of the stock solution of rhodamine phalloidin was prepared using PBS. The RPEs were fixed for 10 min in 4% PFA. The fixed cells were rinsed three times for 5 min each in PBS. The cells were permeabilized for 5 min in the 0.5% Triton X-100 solution. The fixed cells were rinsed three times with PBS. The cells were stained with rhodamine phalloidin for 30 min at RT in the dark. The cells were rinsed three times in PBS for 5 min each time. On day 2, the sections were washed with PBS five times; they were cultured with goat anti-mouse IgG (H + L) highly cross-adsorbed secondary antibodies, Alexa Fluor 488 (Abcam, ab150117), and rhodamine phalloidin (Invitrogen, R415, Waltham, MA, USA) in the dark for 2 h at RT and then mounted with aqueous DAPI (Abcam, ab104139) for confocal imaging (Zeiss, LSM900, Oberkochen, Germany).

### 2.13. Statistical Analysis

Comparisons between the different groups were made using ANOVA or Student’s *t-*test. A *p*-value < 0.05 was taken as statistically significant. All the data are presented as the means ± standard deviations (SDs). The statistical analysis was conducted using SPSS version 22.0 (accessed on 20 August 2022).

## 3. Results

### 3.1. Clinical Characteristics

The 26-year-old proband (Figure 1A, II:3) had blurred vision without photophobia or night blindness for 13 years before being referred to our eye hospital. The proband was the third-born son of a nonconsanguineous couple (Figure 1A; I:1 and I:2). The pregnancy was uncomplicated, and the average birth weight and length were standard in pregnancy at 38 weeks. There was no other reported neonatal problem. An elder brother had glioblastoma and died at 30 years old (Figure 1A; II:1). Unfortunately, no ocular examination or genetic studies were performed on his brother.

The right BCVA of the proband was 0.4, and the left BCVA was 0.03. The slit-lamp result was unremarkable. The intraocular pressure was 13 mmHg in the right eye and 15 mmHg in the left eye. The right-eye axial length was 23.8 mm, and the left-eye axial length was 25.5 mm. He showed myopia and anisometropia, with −4.25 D and −9.25 D equivalent spheres corresponding to the right and left eyes. Bilateral atrophy was recognized in the macular region upon fundus inspection. It was about three disc diameters of the right eye and five disc diameters of the left eye. The atrophy presented total chorioretinal maldevelopment with well-circumscribed boundaries. It could be ophthalmoscopically visualized as the bare sclera with a white appearance due to the lack of choroidal vessels and retinal structures. It exhibited increased backscatter around the lesion for the naked sclera (Figure 2A). In OCT, a crater-like depression of the atrophy in the macula was observed. There were sharp margins and significant excavations at the foveal region of both eyes, accompanied by the absence of retinal pigment epithelium (RPE) and choroid in the lesion (Figure 2B). The toxoplasma, HIV, herpes, syphilis, and rubella serology tests all returned negative results. On the basis of his ophthalmic presentations, he was diagnosed with bilateral macular coloboma.

### 3.2. Mutation Screening and Pathogenic Analysis

Using WES, the compound heterozygous variants c.5255C>G (p.Ser1752Ter) and c.4301delA (p.Asp1434fs*3) were detected in exons 39 and 31 of the *MPDZ* gene of the proband (Figure 1A; II:3) and segregated by disease status. These two variants were later confirmed in the proband’s parents (Figure 1A; I:1 and I:2) and elder brother (Figure 1A; II:2) by Sanger sequencing (Figure 1B). The heterozygous nonsense variant of c.5255C>G was found in the unaffected father (Figure 1A; I:1), and the heterozygous frameshift variant of c.4301delA was found in the unaffected mother (Figure 1A; I:2) and unaffected elder brother (Figure 1A; II:2) (PP1_Supporting).

The *MPDZ* gene has 47 coding exons; the nonsense variant c.5255C>G on exon 39 was predicted to produce an immature termination codon at residue 1752. The variation of Ser1752Ter (TCA changed to TGA) includes guanine nucleotides instead of cytosine nucleotides, causing a 1752 stop codon (TGA) premature termination. The variant of c.5255C>G has been reported in a heterozygous state to be associated with the RP phenotype [8]. Frameshift variant c.4301delA (p.Asp1434fs*3) has not been previously reported in the literature and is not found in the dbSNP, ExAC, gnomAD, 1KGP (including 301 Chinese), or the exome intensive consortium database (including all races, male: 33644, female: 27062, total: 60706), and the mutational frequency of this variant was equal to 0 (PM2_Supporting). The premature termination codon (PTC) occurs in exon 31 with Asp1434fs*3 (GAT GCA GTG AAT… changed to GTG CAG TGA AT). The heterozygous variation at 4301nt of Asp1434fs*3 includes one nucleotide deletion of exon 31, causing a frame-shift variation, which makes the following three codons change to Val 1434 (GTG), Gln 1435 (CAG), and stop codon 1436 (TGA) from Asp 1434 (GAT), Ala 1435 (GCA), and Val 1436 (GTG). The frameshift variation leads to neither alternative splicing nor exon skipping.

The truncate mutations would result in the premature termination of polypeptide chain synthesis, and most of the proteins produced were inactive or lost their normal function (PVS1_Very Strong). Nonsense-mediated mRNA decay (NMD) is expected to eliminate MPDZ protein expression (including truncated polypeptides) from both mutant alleles.

NCMD is a developmental abnormality affecting the macula and may be associated with significant visual impairment, which is similar in phenotype [19]. Heterozygous, noncoding variations upstream of the PRDM13 gene were reported to be associated with NCMD; therefore, the affected individual also underwent a Sanger sequence to elucidate the relevant remote variants as the cause of macular coloboma. No other relevant variations were detected.

In summary, the novel compound heterozygous variations c.5255C>G (p.Ser1752Ter) and c.4301delA (p.Asp1434fs*3) of the *MPDZ* gene are probably the pathogenic variants for this MC family according to the ACMG guidelines. This is because

(1)The two variants were observed in the database of normal subjects of the sequence company with a very low mutation frequency of 0;(2)Other potential virulent variants were not detected in this study;(3)Both the nonsense variant p.Ser1752Ter and the frameshift variant p.Asp1434fs*3 were nonfunctional variations and were considered solid pathogenic evidence;(4)There was a sufficient correlation between the gene variations and MC phenotype.

### 3.3. MPDZ Knockdown Results in Cytoskeleton Rearrangement and Enhanced Motility

To determine whether or not the deletion of MPDZ could change the stability of human RPE, we knocked down *MPDZ* with siRNA in cultured human RPE. Furthermore, siRNA-mediated knockdown and lower MPDZ expression were confirmed by Western blotting. MPDZ inhibition resulted in cytoskeleton rearrangement and reduced the ZO-1 expression (Figure 3B). In addition, MPDZ inhibition activated RhoA but not Rac1 and RPE migration. These results suggest that MPDZ is involved in maintaining RPE structure and stability.

### 3.4. Expression of MPDZ in OLM of Zebrafish

Whole-mount in situ hybridization (WISH) was performed using zebrafish larvae (more than 20 in each group) at 36 h post-fertilization (hpf) to determine whether the identified variation would alter the stability of the retina. As a result, it was proven that MPDZ was strongly expressed in the zebrafish eyes. In order to further determine the expression pattern of MPDZ in the eyes of zebrafish, IF staining was carried out in sections. MPDZ was observed to be mainly expressed in the ellipsoid zone of the zebrafish. Following the knockdown of MPDZ, the ZO-1 was less expressed at the OLM, indicating the critical role of MPDZ in the retina of zebrafish.

### 3.5. Developmental Defects in MPDZ Crispants Zebrafish

To better determine the pathogenesis and explain the phenotypic diversity caused by the *MPDZ* variation, we investigated the loss of function of the MPDZ in zebrafish. *MPDZ* crispants were generated using the CRISPR/Cas9 system. Furthermore, IF staining was performed to validate the knockdown efficiency in *MPDZ* crispants, and the result demonstrated a dramatic inhibition of MPDZ and ZO-1 expression in the *MPDZ* crispants compared with that in wildtype zebrafish. Notably, the *MPDZ* crispants zebrafish displayed reduced eye area and body length compared to the wildtype zebrafish (Figure 4A). Histological examination showed a series of phenotypes, including a small eye and a thin retina (Figure 4C). To confirm the localization of MPDZ using immunofluorescence and confocal microscopy, we observed intense immunostaining at the ellipsoid zone of the retina, proving MPDZ localization (Figure 3A Scramble gRNAs + Cas9 group, yellow arrow), while the ZO-1 was mainly at the outer limiting membrane (OLM) of the zebrafish (Figure 3A Scramble gRNAs + Cas9 group, blue arrow). For the knockdown of MPDZ, a less intense immunostaining at the ellipsoid zone of the retina (Figure 3A; *MPDZ* gRNAs + Cas9 group, yellow arrow) and outer limiting membrane (OLM) (Figure 3A; *MPDZ* gRNAs + Cas9 group, blue arrow) was detected.

## 4. Discussion

### 4.1. MPDZ Variation and Influence

As previously reported in the literature, MPDZ, also known as MUPP1, is mainly localized on autosome 9p23, with a size of 174,397 bp. It contains 47 exons and encodes the MPDZ protein. Most *MPDZ* gene variations are missense variations, while a few are splice site, frameshift, and synonymous variations. To date, 162 variations have been reported to be likely pathogenic or globally pathogenic (Clinvar database). The pathogenic site variations with different phenotypes are presented in Figure 5. MPDZ is an acronym derived from multiple PSD95, DLG, and ZO-1; it contains 13 PDZ domains, which are characteristically involved in multiple protein–protein interactions. MPDZ acts as a scaffold protein for tight-junction-related proteins, adherens-junction proteins, and transmembrane receptors [23]. It has been shown that the PDZ domains of some PDZ-containing proteins interact with the specific ion channel subunits of C-terminal tail sequences and G-protein-coupled receptors. MPDZ regulates cell processes such as cytoskeletal organization, cell polarity, cell proliferation, and many signaling pathways through its PDZ domains [24,25]. The MPDZ protein can participate in many possible interactions due to its multidomain structure. Each domain is highly conserved across species, and each is made up of six beta-sheets and two alpha-helices.

In the present study, the novel compound heterozygous variations c.4301delA (p.Asp1434fs*3) and c.5255C>G (p.Ser1752Ter) in the *MPDZ* gene were identified to be responsible for autosomal recessive macular coloboma. The first variant, c.4301delA, was predicted to cause a change in the coding framework at codon 1434 and alter the amino-acid sequence leading to the truncation of peptide synthesis. The second nonsense variant, c.5255C>G, was predicted to generate a premature termination codon at residue 1752, resulting in a truncation of the protein. According to ACMG, both the nonsense variant p.Ser1752Ter and the frameshift variant p.Asp1434fs*3 were nonfunctional variations and classified as pathogenic variants. The compound heterozygous variation probably led to a complete loss of function of the MPDZ protein. Sanger sequencing further proved that this compound heterozygous variation of *MPDZ* existed in the proband of this MC family, whereas the family members (I:1, I:2, and II:2) with only one heterozygote showed no symptoms. We did not detect the second gene responsible for pathogenetic retinal dystrophy in this family through WES. Therefore, the variations of c.4301delA and c.5255C>G A in the *MPDZ* gene were presumed to be autosomal recessive pathogenic variations of MC.

### 4.2. MPDZ Location and Function in the Retina

MPDZ is found in the central nervous system, retina, and many other tissues, and it has been shown to be localized in tight, gap, neuromuscular, synaptic, and adherens junctions [26,27,28,29]. *MPDZ* gene variants have been reported to be associated with congenital hydrocephalus. The disruption of intercellular junctions in the ventricular region might be a common cause of this disease [10,30]. Moreover, a previous report on compound heterozygous variants (p.Gly132Ser and p.Leu582Val) of the *MPDZ* gene also highlighted an association with the congenital communicating hydrocephalus phenotype [9]. In the retina, only heterozygous *MPDZ* variants have been observed in RP and LCA. MPDZ was previously reported in the adherens junction (AJ) of the OLM. This is a special interaction region between the inner segments of the photoreceptor and the innermost process of the radial Müller glia [8]. Our study found that MPDZ was highly expressed in the photoreceptor inner segments, while it was accompanied by ZO-1 expression at the OLM layer of the wildtype zebrafish. These findings are consistent with previous reports about MPDZ’s location in humans and mice.

In order to further evaluate the *MPDZ* variation, experiments were carried out to analyze the effect of MPDZ knockdown on the zebrafish model. Histological sections were analyzed to confirm that the absence of a full-length MPDZ protein caused an abnormal phenotype of the eye and brain. Various techniques have realized success in studying aspects of the development of retinopathy. According to our results, the variations led to no protein expression (null alleles) and dramatically destroyed the OLM’s function. ZO-1 was found to degenerate following MPDZ loss, accompanied by a severely thinner OLM. Our study revealed that knockdown of the zebrafish MPDZ caused a decrease in the ellipsoid zone, destroyed the outer limiting membrane, and could cause subsequent RPE degeneration. The *MPDZ* variations caused a decrease in the MPDZ protein expression and also suspended the TJ protein assembly and stability. The accompanying reduced ZO-1 protein is a core component of the MPDZ complex, and MPDZ is also important in maintaining regular retinal function. TJs, together with AJs and desmosomes, which are parts of the epithelial and endothelial junctional complex, are involved in the blood–brain and blood–retina barriers [12,31]. Pharmacological interference or genetic disruption of the components of the AJ and/or the supporting scaffold most likely alters the distribution of the TJs, which leads to significant impairment in OLM integrity [32,33]. Crb1, MPP4, PALS1, and aPKC, located in the AJs, are reported to be essential for photoreceptor morphogenesis and retinal degeneration [33,34,35].

In this study, we discovered that the tight junction (TJ) protein MPDZ is associated with incomplete retina differentiation. This accords with previous studies reporting that another TJ-related protein, claudin-19, is also related to the MC phenotype. Altogether, this suggests that TJs located on the OLM correlate with human macula development, and disruption of the protein may result in the MC phenotype. A potential explanation for the diverse genotype–phenotype correlations of TJ genes is a combination of their multifunctionality and the various negative effects of their mutant alleles. Further confirmation through the in vivo study and screening of TJ-related variations in more MC patients is necessary.

### 4.3. MC-Related Gene Variations

Using WES, we identified the novel compound heterozygous *MPDZ* variation, c.5255C>G (p.Ser1752Ter) and c.4301delA (p.Asp1434fs*3), responsible for an autosomal recessive macular coloboma in a Chinese family. The affected proband had isolated bilateral MC without systemic abnormalities. MC was first recorded in 1935 as an inherited congenital disease characterized by incomplete differentiation of the arcuate bundles along the horizontal raphe during embryonic development [6]. Unfortunately, few genes have been determined to be related to MC to date. MC occasionally presents in some genetic disorders, such as Down’s syndrome and FHHNC (familial hypomagnesemia with hypercalciuria and nephrocalcinosis), or is coincident with retinal dysplasia and degeneration (*rdd*) [36,37]. MC was recorded as appearing in *CLDN-19*-gene-related FHHNC and *CNNM4-*gene-related Jalili syndrome (JS). LCA5 and RDH12 are associated with Leber congenital amaurosis (LCA), and MC is one of the phenotypes for LCA5-LCA and RDH12-LCA. The *DHX38* gene is associated with early onset retinitis pigmentosa (eoRP), and MC was discovered in DHX38-eoRP. Moreover, variations in *BEST1* and *RIMS1* were reported in a Chinese patient with bilateral macular coloboma [7]. According to previous reports, variations in genes known to be responsible for retinal degeneration were found to be associated with MC, highlighting that some severe congenital disabilities could be associated with multiple variations in multiple genes, making gene therapy more challenging.

In this study, it was identified that the encephalopathy formation gene causes MC, in contrast with the prevailing pattern observed in hydrocephalus. Other cases have been reported that support the correlation between cerebral dysplasia and ocular defects. Although a 5-year-old male was reported to have hydrocephalus and chorioretinal atrophy, genetic diagnosis was, unfortunately, not performed [2]. A compound heterozygous splicing *CDON* variant was identified in patients affected with bilateral coloboma of the iris, retina, and choroid and was also reported in holoprosencephaly (HPE) and pituitary stalk interruption syndrome [38]. These disorders are manifestations of the link between cerebral and retinal development. We propose that the ellipsoid zone of the retina plays a critical role in correlation with the cerebrum.

### 4.4. The Phenotype Varied According to Previously Reported Cases

Previous human *MPDZ* variations have been associated with congenital hydrocephalus (HYC2), severe brain malformations, and eye phenotypes limited to iris coloboma (OMIM *603785, #615219), with cardiac septal defects or nonspecific dysmorphic features in some cases. In our study, the compound heterozygous variation of *MPDZ*, comprising the nonsense variant Ser1752Ter (in exon 39) and the frameshift variation Asp1434fs*3 (in exon 31) as null alleles, was identified in the proband of a Chinese family, and the phenotype was characterized by isolated macular coloboma in the eyes but with no brain malformation or hydrocephalus. Both variations could cause a premature termination codon (PTC) leading to truncation of the protein and in the dysfunction of the truncated protein. Generally, the truncated protein in truncated mutations could give to rise a toxic gain function, and the mutant allele could exert a dominant negative effect on the wildtype allele, aggravating the haploinsufficiency of the protein, which is not enough to maintain normal function. All this could cause a severe phenotype in principle. However, the proband only showed a mild phenotype. In reviewing previous reports on HYC2 related to *MPDZ* truncation variations, such as homozygous variants of Gln1490Argfs*19 and Gln210*, as well as compound heterozygous variants Arg744* and Arg1071*, we found some cases of a relatively milder phenotype in HYC2 despite an absence or low presence of the MPDZ protein [10,11]. It is worth noting that some HYC2 cases with MPDZ truncation variations also presented iris defects and retinal abnormalities, but the proband in our study only showed isolated macular coloboma without HYC2.

Although the mechanism for phenotypic diversity in these cases remains unclear, it was first speculated that the variability is caused by modifying genes and epigenetic and environmental factors or that it is related to differences in splicing factor machinery. Then, it was observed that the truncated variations found in heterozygosity contribute to a milder phenotype. As in the case of the non-affected parents, the wildtype allele would produce enough wildtype protein to counterbalance the negative effect of the preserved fraction of the mutant transcripts. Last but not least, NMD is believed to play a key role in the inconsistency in the association of the functional consequences with the genetic variation in the cases studied. The nonsense and frameshift variations in our case generate a premature stop codon and C-terminal truncation variations in MPDZ. This could introduce PTCs which can be recognized and degraded by NMD. NMD works as an RNA surveillance system of mRNA quality that widely exists in eukaryotic cells. It can recognize the mutant mRNAs carrying PTCs and rapidly degrade and eliminate aberrant transcripts to prevent the accumulation of truncated and potentially harmful proteins by preventing the translation of the aberrant mRNA, thereby modifying the phenotypes.

NMD is efficient with PTCs located upstream of the last exon junction complex (EJC) [39]. This efficiency is governed by several rules which were consistent with our data. The 50nt rule and the last exon rule are taken as canonical rules [40,41,42]. PTCs of less than 50 nucleotides (nt) upstream of the last exon–exon junction will trigger NMD (the 50nt rule), and PTCs in the last exon of a gene do not trigger NMD (the last exon rule). Our study involved two variations, Asp1434fs*3 and Ser1752Ter, located on exon 31 and 39 in *MPDZ*, respectively. As is consistent with the existing rules, the physical sites of these two variants were not on the last or the penultimate exon as NMD is believed to play a role on the phenotypes. Additionally, our data also support the long exon rule and the start-proximal rule, which are taken as “non-canonical rules” [43,44]. These rules state that long exons (greater than approximately 400nt) inhibit NMD and that PTCs less than 150nt from the start codon fail to trigger NMD. Exon 31 contains 91nt, and exon 39 contains 149nt; they are located in the neither proximal part nor the border of the intron of the *MPDZ* gene with 47 exons. Thus, the two variations are predicted to efficiently trigger NMD. In addition to the canonical EJC-dependent NMD, an alternative EJC-independent NMD targets mRNAs with an aberrant architecture downstream of the termination codon, such as an unusually long 3′ untranslated region (UTR), and this was also observed in our study of *MPDZ* mutation [45].

Although MPDZ is reported to be associated with other *rdd* diseases such as RP and LCA, MC is another phenotype representing a congenital disorder of the macular region and choroid. MC may present in different forms in other diseases such as Best vitelliform macular dystrophy (BVMD), advanced cone–rod dystrophy (CORD), congenital toxoplasmosis macular scarring, Leber congenital amaurosis, and central areolar choroidal dystrophy [7].

In this study, typical bilateral MC symptoms were identified, including severe vision loss and cup-shaped lesions with a complete lack of retina and choroid. We extend the ocular phenotypic spectrum for MPDZ-associated *rdd*, emphasizing its vital role in the retinal development of MPDZ and the promising role of WES in revealing the novel disease-causing genes associated with monogenic diseases.

## 5. Conclusions

In this study, we confirmed the role of MPDZ in autosomal recessive human MC through the identification of the novel compound heterozygous variation (p.Asp1434fs*3 and p.Ser1752Ter) of the *MPDZ* gene in a Chinese family. Taking together the genetic findings and the phenotypes of the corresponding zebrafish models, the novel compound heterozygous variation of the *MPDZ* gene was evidenced to cause macular coloboma via both recessive negative and loss-of-function effects, respectively. NMD is hypothesized to play a key role in the inconsistency of the phenotype in our case. The co-occurrence of hydrocephalus and coloboma and the overlapping genetic etiology between the two conditions suggest that similar pathways are involved in both brain and macular development. The screening of *MPDZ* in additional MC and hydrocephalus patients for an evaluation of the genotype–phenotype correlations is needed to discover its etiology. Furthermore, given that the OLM is essential in macular development, it is suggested that disruption of the OLM may increase the odds of congenital MC.

## Figures and Tables

**Figure 1 cells-11-03602-f001:**
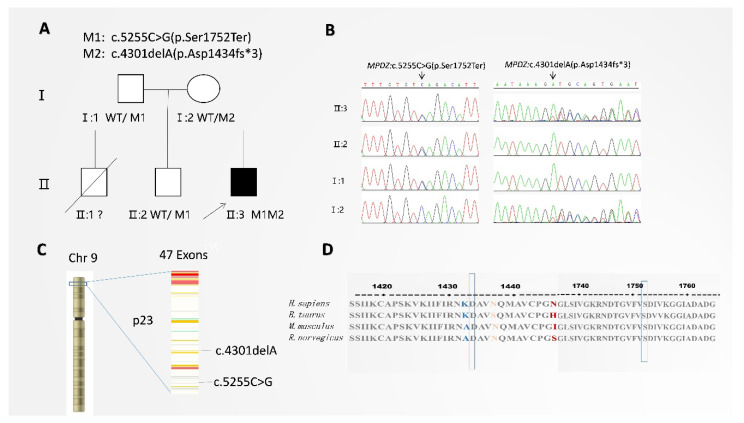
Validation of the mutation. (**A**) Pedigree information. In this pedigree, filled symbols indicate affected patients with MC. The unfilled symbols indicate unaffected individuals. An oblique line indicates a deceased person. Squares denote males, and circles denote females. M1 represents the c.5255C>G variant (p.Ser1752Ter), and M2 represents the c.4301delA variant (p.Asp1434fs*3). Proband (II:3) is indicated by an arrow. It shows a novel compound heterozygous *MPDZ* variation, c.4301delA (p.Asp1434fs*3) and c.5255C>G (p.Ser1752Ter). (**B**) Sanger sequencing identifying the breakpoints at c.4301delA (p.Asp1434fs*3) and c.5255C>G (p.Ser1752Ter). (**C**) *MPDZ* is localized on autosome 9p23 and contains 47 exons. (**D**) Arginine at positions 1752 and 1434 is highly conserved among different species according to proteomic conservation analysis.

**Figure 2 cells-11-03602-f002:**
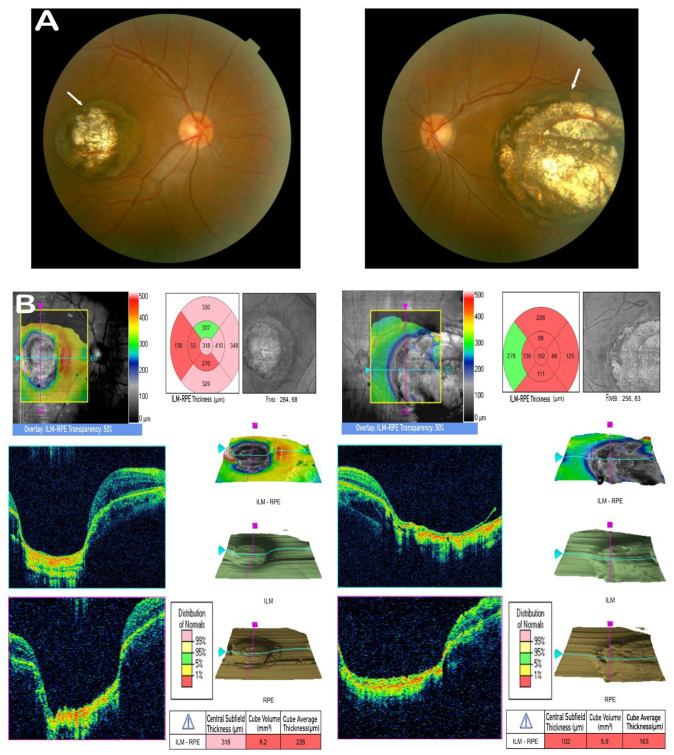
Clinical observations. (**A**) Fundus photograph of the proband (II:3) exhibiting prominent bilateral atrophy in the macula (white arrows); the atrophy presents well-circumscribed borders, total chorioretinal maldevelopment with a white appearance, and increased backscatter around the lesion due to the bare sclera. (**B**) OCT scan of the proband (II:3) showing a cup-shaped lesion with the complete absence of retina and choroid. There is a sharp margin, with significant excavation in the foveal region of both eyes.

**Figure 3 cells-11-03602-f003:**
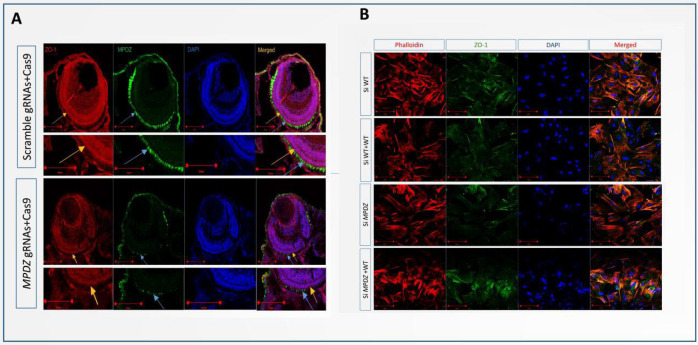
(**A**) IF staining showing that MPDZ localization in the ellipsoid zone of the MPDZ crispants retina, with intense green immunostaining (blue arrow), was decreased compared to the wildtype. ZO-1, mainly at the outer limiting membrane (OLM) (yellow arrow), was less expressed following the knockdown of MPDZ. (**B**) MPDZ knockdown with siRNA in cultured human RPE. IF staining showing the inhibition of MPDZ leading to cytoskeleton rearrangement and reduced ZO-1 expression.

**Figure 4 cells-11-03602-f004:**
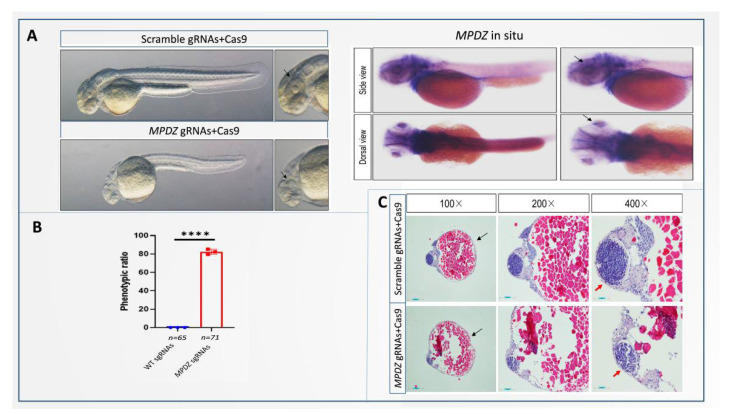
Experiment on zebrafish. (**A**) *MPDZ* crispant zebrafish displayed reduced eye area and body length compared to wildtype zebrafish. The phenotype of the *MPDZ*-knockdown crispants was more obvious compared to that of the wildtype. (**B**) Phenotypic ratio indicating the percentage of fish that uniformly showed reduced eye area and body length. The percentage of phenotypic positive fish in the *MPDZ* gRNA group (*n* = 71) was significantly greater than that in the WT gRNA group (*n* = 65) (*p* < 0.05). (**C**) Histological examination showing a series of phenotypes, including a small eye and a thin retina.

**Figure 5 cells-11-03602-f005:**
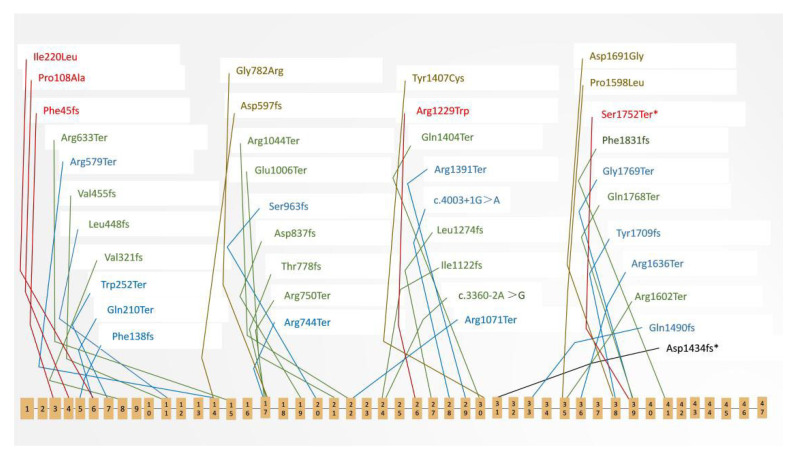
*MPDZ* pathogenic variations in humans. Schematic representation of the human MPDZ protein showing the distribution of variations. Blue represents a variation related to the hydrocephalus phenotype in Clinvar (https://www.ncbi.nlm.nih.gov/clinvar/, accessed on 20 August 2022). Green indicates that the pathogenic condition of the variation was “not provided” in Clinvar. Red and brown indicate that the variation was reported with the phenotype of retinitis pigmentosa (RP) and Leber congenital amaurosis (LCA), respectively. Asterisks denote variants related to macular coloboma (MC) in the present study.

**Table 1 cells-11-03602-t001:** Primers used for *MPDZ*-Oligos.

Name	Sequence of Oligonucleotide (5′–3′)
Scaffold Primer	AAAAGCACCGACTCGGTGCCACTTTTTCAAGTTGATAACGGACTAGCCTTATTTTAACTTGCTATTTCTAGCTCTAAAAC
*MPDZ*-oligo-1	TAATACGACTCACTATAGGCTTTCTGCAGCTGTGGACGTTTTAGAGCTAGAAATAGC
*MPDZ*-oligo-2	TAATACGACTCACTATAGGCACGCTGGCCGCACGCTCGTTTTAGAGCTAGAAATAGC
*MPDZ*-oligo-3	TAATACGACTCACTATAGGATTAAAGGTAATGCCGAGGTTTTAGAGCTAGAAATAGC
*MPDZ*-oligo-4	TAATACGACTCACTATAGGACGTGGCTCCGCCGGGCTGTTTTAGAGCTAGAAATAGC
*MPDZ* probe F	CGACGAGCTGTTGGAGATAAA
*MPDZ* probe R	TTCCCGCCGACTATACTAAGA

## Data Availability

Not applicable.

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
