# Peer review of "Novel Compound Heterozygous Variations in MPDZ Gene Caused Isolated Bilateral Macular Coloboma in a Chinese Family"

_cells, 2022, doi:10.3390/cells11223602_

Round 1
Reviewer 1 Report
This manuscript by Shuang Zhang et al. reports biallelic MPDZ mutations (multiple PDZ domain protein, crumbs cell polarity complex component) in a family with isolated bilateral macular coloboma. This new association is a potentially useful contribution to the clinical genetics of eye disease. However, the writing is poor – and misleading in many places.
Previous human MPDZ mutations have been associated with congenital hydrocephalus (HYC2), severe brain malformations and eye phenotypes limited to iris coloboma (OMIM 603785, 615219) -- with cardiac septal defects or nonspecific dysmorphic features in some cases.
Heterozygous MPDZ mutations were also reported in 10/276 unsolved LCA and RP cases, including 3 null alleles (Ali et al. IOVS 2011) – but their significance is unknown. In principle, these cases may harbor undetected second MPDZ mutations or unlinked mutations in interacting genes (digenic RP).
The MPDZ protein is 2070 amino acids in length. It includes 3 disordered regions and 13 PDZ domains. Published alleles include Q217X, Q1490RfsX19, R744X and R1071X – which are predicted to trigger NMD (nonsense-mediated mRNA decay), with no protein expression (null alleles) – and missense allele A1760T, which was not functionally evaluated.
Major comments
1. The new mutations, S1752X (in exon 39 of 48) and D1434fsX2, should also cause NMD, and thus act as null alleles, yet no brain malformation or hydrocephalus is reported. Why is the phenotype milder in this family? More discussion on this point is warranted.
The authors should give the number of nonsense (out-of-frame) codons predicted to occur between Asp1434 and the stop codon, as D1434fsX2 (GAT CCA GTG A … changed to GTC CAG TGA).
The authors should fully discuss nonsense-mediated mRNA decay (NMD), which is expected a priori to completely eliminate MPDZ protein expression (including truncated polypeptides) from both mutant alleles. The gene has 47 coding exons, and the PTC (premature termination codons) occur in exons 31 and 39 (line 204 in the Results). Consequently, the lengthy discussion regarding pathogenicity of specific truncated protein forms – and the zebrafish rescue experiments with truncated MPDZ forms (Fig 4) – are probably moot (pointless). Logically, if there is no mRNA, then there will be no protein.
Here, the authors should use common sense, rather than a blizzard of ClinVar coding and ACMG guideline jargon, to establish the pathogenicity of variants. Given NMD, the entire discussion regarding protein effects (starting at line 215) is irrelevant and should be removed. Likewise, the molecular modeling diagrams are meaningless distractions (Fig. 2).
However, if specific, shorter MPDZ isoforms are known to arise via alternative splicing, S1752X mRNA might escape NMD. The authors might wish to consider this point.
Do either S1752X or D1434fsX2 also generate new splice donor sites? If so, this may allow mRNA splicing to occur, restoring the translational reading frame and causing a small internal peptide deletion – with preservation of some MPDZ function. The authors could consider this explanation for the comparatively mild phenotype in the proband. If so, then the zebrafish and RPE cell experiments could be performed with more relevant, internally deleted isoforms.
For more details on NMD, the authors should read:
Holbrook et al. Nonsense-mediated decay approaches the clinic. Nat Genet 36, 801–808 (2004).
Khajavi et al. Nonsense-mediated mRNA decay modulates clinical outcome of genetic disease. Eur J Hum Genet 14:1074-81 (2006) PMID: 16757948.
Dyle et al. How to get away with nonsense: Mechanisms and consequences of escape from nonsense-mediated RNA decay. WIREs RNA 11:e1560 (2020).
2. Causation. The pedigree has one affected and one heterozygous child. The mutations were identified by WES and confirmed by Sanger sequencing. Given the novel clinical findings for MPDZ mutations, the authors should rule-out other known genetic causes for macular coloboma, including noncoding dominant mutations in the remote enhancer for PRDM13, associated with clinically similar North Carolina Macular Dystrophy (MCDR1), which would not be detected by WES (OMIM 136550).
3. Potential role for MPDZ mutations in RP and LCA. Please expand and qualify this phrase “c.5255C>G has been reported in the HGMD database (HGMD CM117304) to be associated with retinitis pigmentosa (PS1_Strong)”. The variant has ONLY been observed in a heterozygous state in one RP case (line 205 in Results).
In the Discussion, distinguish mutations reported in affected patients (with likely causation) from database variants recorded in normal individuals (mostly neutral).
Only heterozygous MPDZ variants have been observed in RP and LCA, so no causation was established. Therefore, the Discussion sentence beginning in line 374 is wrong. (“MPDZ has also been reported to be responsible for degenerate retinal diseases such as RP and LCA.”)
Discuss retinal findings in the parents and brother, who are heterozygous for MPDZ truncating mutations, similar to cases reported by Ali et al. (2011). Do they have impaired retinal sensitivity or ERGs?
Given the function of MPDZ, were other, peripheral retinal defects observed in the proband – retinoschisis for example? Are scotopic ERGs preserved?
3. The manuscript has numerous grammatical and stylistic errors, which must be corrected for publication – well beyond the list detailed in “Minor comments”
Fig. 3 is very cluttered – please redraw this diagram in a conventional, informative way.
Minor comments
1. Change ‘biocular’ to ‘bilateral’ (3 instances, including title)
2. line 45-46 italicize all gene names (BEST1, RIMS1, and CLDN19)
3. line 49-50 change “similar mutations in MPDZ were identified” to “heterozygous mutations in MPDZ were identified”
4. line 53-56 cite work on Mpdz KO mice and ependymal cell integrity (Feldner et al. Loss of Mpdz impairs ependymal cell integrity leading to perinatal-onset hydrocephalus in mice. 2017, EMBO Mol Med 9:890-905. PMID: 28500065)
5. line 60-63 change “compound heterozygous mutations could lead to developmental defects in the retina, with the aid of recessive-negative and loss-of-function effects, respectively” to “compound heterozygous loss-of-function mutations could lead to developmental defects in the retina”
6. line 86 change “After low-quality reading,” to “After filtering low-quality reads,”
7. line 87-88 change “We detected the insertion and deletion realignment, quality recalibration, and variant calling from GATK” to “We performed insertion and deletion realignment, quality recalibration, and variant calling using GATK”
8. line 170 change “underwent” to “had”
9. line 174 change “Family history, one of his elder brothers had glioblastoma” to “An elder brother had glioblastoma”
10. line 177 change “The Slit-lamp did not show remarkable alteration in both eyes” to “The Slit-lamp was unremarkable”
11. line 177-178 too much precision for IOP. Use 13 and 15 mm Hg (without decimal). Likewise for axial length – you cannot measure length with 10-micron accuracy!
12. line 195 change “were initially detected in exons 39 and 31 of the MPDZ gene of the proband (Fig.1A II:3) and then segregated the disease status” to “were detected in exons 39 and 31 of the MPDZ gene of the proband (Fig.1A II:3) and segregated with disease status”
13. line 203-204 discuss nonsense-mediated mRNA decay (NMD), which is expected to eliminate MPDZ protein expression (including truncated polypeptides) from both mutant alleles. The gene has 47 coding exons, and the PTC (premature termination codons) occur in exons 31 and 39.
14. line 208 correct “common people database of gene sequencing companies” to a real database name.
15. line 249-253 and elsewhere – should be “CRISPRants”
16. line 250 spell out “immunofluorescence”
17. line 258 should be “outer limiting membrane (OLM)”
18. Fig 1C – MDPZ gene is inverted in the chr 9p23 diagram (transcribed centromere to telomere)
19. line 297-299 cite the source for variants shown as “The green represents the pathogenic condition of the variation that was not provided by ClinVar. The red and the brown represent the variation that was reported with the phenotype of retinitis pigmentosa (RP) and Leber Congenital Amaurosis (LCA), respectively.”
20. Eliminate Fig 4C – the glioblastoma lesion in II-1 is unrelated to the MPDZ gene mutations, and the genotype of this individual is not known. Likewise, remove this speculative part of the Discussion (lines 450-458).
21. graphs in Fig 4 are unreadable – the size is way too small
22. line 430 rephrase “dominant pattern observed in hydrocephalus” to avoid confusion with genetic dominance. Do you mean “prevailing”?
23. line 55 spell out CSF (cerebral spinal fluid)
24. line 76-78 what contains 100 health care from the same ethic group? This section is confusing what are the authors are trying to say?
25. line 90 “healthy population frequency databases” is awkward
26. line 119-121, line 161-162, and line 165-166 poorly written. Rephrase.
27. line 136 “one cell stage” and 144 “1-cell stage” formats are inconsistent.
28. line 139-140 change “normal MPDZ” to “wild-type MPDZ”
29. line 187-189 The authors mention that retina and choroid were completely lacking at the macula, but that abnormally thin retina was observed at the foveal region – this seems contradictory
30. line 208 what does “common people database” mean?
31. line 209-211 change “1000 genomic database” to 1KGP database”
32. line 212-214 unclear writing – and irrelevant, as noted above
33. line 221 change “functionss” to “functions”
34. line 237 “this phenotype” – what does this refer to, the MC phenotype?
35. line 244 “normal human MPDZ” – use “wild-type” instead
36. line 253-258 how do “control” and “wild-type” zebrafish differ?
37. line 253 “reduced eye area” – how was this measured?
38. line 261 and 264 “normal humans” – awkward phrasing
39. line 262 “phenotypic ratio” – what does this refer to? In the rescue experiment, it is not clear How variable the phenotypes are within and between models (wt or mut mRNA injection)? What age embryos were injected? Provide tables listing the number of embryos screened by histology for each model and how frequently each of the described phenotypes was observed.
40. line 269-274 Show wholemount ISH results. Justify choosing 36 hpf for scoring. Is IF staining shown in Fig 5C? If so, indicated this in the text.
41. Fig. 2AB – what do colors indicate?
42. Fig 5 is poorly composed.
Fig 5A
Panel on the top: including a scale bar would be helpful
Bar graph: what is phenotypic ratio?
Panel on the bottom: unequal magnification among panels. Label MPDZ knockdown vs. control. What method was used for staining? What do the arrows show? It is hard to see differences between left and right panels.
Fig 5B add scale bar
Fig 5D Bar graph – what is phenotypic ratio? Define siNC and siMPDZ in panel
43. line 334 change “at home and abroad” to “globally”
44. line 385 “mutations prevented …” what two mutations are meant by this phrase?
45. line 390 “decrease and mislocalization of multiple MPDZ proteins” is unclear. Does this mean decreased protein expression? Are only some MPDZ isoforms mislocalized?
46. line 394-395 rephrase this sentence
47. line 416-418 clarify
48. line 425 “MC multiplied the phenotype of rdd” makes no sense.
49. line 431-432 change “the male child of 5-year-old” to “5-year-old male”
Reviewer 2 Report
The article identified a novel gene of mutation in a rare genetic disease macular coloboma. The authors traced the family history and genome analysis to identify MPDZ gene in MC. After successfully identified the mutation, the authors tried to replicate the disease with ex vivo and in vivo studies in order to prove that MPDZ is responsible for MC. However, there are certain imperfections along with the paper.
1. The figure in the manuscript was too small and low in resolution. The labels in each bar graph were very hard to read. For example Figure 5D, what is the phenotype being analyzed?
2. In section 3.3 which is the in vitro study of RPE cell culture, the supplemental figures were not referred nor the supplemental figure legends were provided. It was confusing to read RPE experiments along with results figures.
3. There was no statistical analysis section in method to outline how significance or error bar in the graphs were produced and the number of replicates.
4. MPDZ is associated with tight junction formation of various cell type, RP and Leber congenital amaurosis are all defects of certain cell function. How do authors connect the MPDZ and local tissue coloboma in MC patients? Zebra fish study(Fig5) showed global cell function defect but not local retinal coloboma, how would the author explain the pathological differences between patients’ tissue level defect and cell level defect seen in zebra fish?
Round 2
Reviewer 1 Report
This revised manuscript by Shuang Zhang et al. reporting biallelic MPDZ mutations (multiple PDZ domain protein, crumbs cell polarity complex component) in a family with isolated bilateral macular coloboma is improved. However, it still requires significant scientific and editorial (writing) revisions.
1. The authors have not grasped the concept of “nonsense peptide”. To help communication, this reviewer spelled out the consequences for D1434fsX – truncation of the MPDZ polypeptide after only TWO codons, so the allele should be termed D1434fsX2 throughout the manuscript. The reviewer even parsed the sequence exactly in the comments, so that this would be clear – “Asp1434fsX2 (GAT GCA GTG AAT changed to GTG CAG TGA AT)” Once the TGA stop codon is encountered (3rd codon after D1434), the ribosome disengages, and no more codons are translated. Likewise, the S1752X polypeptide terminates at codon 1752 (TCA changed to TGA).
Therefore, the statement, “There are 318 nonsense codons between Ser1752 and the stop codon, and 634 nonsense (out-of-frame) codons between Asp1434 and the stop codon” in the text (lines 261-265) are wrong and should be deleted.
For deeper understanding of nonsense peptides, the authors are encouraged to read Crick’s original report using frameshift mutations to define the triplet genetic code and Yanofsky’s commentary on this discovery.
Crick FH, Barnett L, Brenner S, Watts-Tobin RJ (1961) General nature of the genetic code for proteins. Nature 192:1227-1232, PMID 13882203.
Yanofsky C (2007) Establishing the triplet nature of the genetic code. Cell 128:815-818, PMID 17350564.
2. The discussion of cryptic splice sites is VERY problematic and needs serious attention. The authors claim to have applied bioinformatic analysis to evaluate the possibility that the two new MPDZ mutations activate cryptic 5’ (donor) splice sites, potentially allowing in-frame translation of (internally) deleted MPDZ polypeptides with partial function. However, no details are provided, and the program they cite (Mutation Tester) is not equipped to perform this type of analysis. So this part of the paper is illogical, if not plainly WRONG.
In particular, the phrase
“This may allow mRNA splicing to occur, restoring the translational reading frame and leading to a small internal peptide deletion that preserve some MPDZ function. The exon containing the stop codon is excluded from the transcript via alternative splicing.”
can only be true if exon 31 contains a multiple of 3 nucleotides – and it does not. (There are 91 nt). Moreover, details regarding the “small internal peptide” would need to be discussed – the amino acid endpoints, relevant MPDZ domain, and specific functional consequences for MPDZ protein function, for example. In other words, the authors would need to fully support their claim that D1434fsX2 creates a hypomorphic allele. This is highly unlikely. Logically, a point mutation in the exon 31 body should not cause exon skipping. There are MANY conceptual errors here.
In fact, when this reviewer applied a simple test of cryptic splicing using the Berkeley Genome project NNSPLICE server (https://www.fruitfly.org/seq_tools/splice.html) to detect new splice donor sites in mutated sequence from exon 31 (D1434fsX2) and exon 39 (S1752X), none were found.
Newer software, to analyze mutant sequences for altered splice sites (such as MaxEntScan, SpliceAI or VarSEAK), could also be applied – if well documented in the manuscript, but I suspect these will confirm that neither mutation alters the MPDZ mRNA splicing – certainly not in ways that would create an in-frame deletion and hypomorphic allele. These in silico analyses should be extensively described in the revised text – and any new splice sites leading to a deleted in-frame protein should be tested functionally, using an exon trapping (Buckler et al. 1991) or splicing reporter minigene assay (Gaildrat et al. 2010).
Buckler AJ, Chang DD, Graw SL, Brook JD, Haber DA, Sharp PA, Housman DE (1991) Exon amplification: a strategy to isolate mammalian genes based on RNA splicing. Proc Natl Acad Sci USA 88:4005-4009, PMID 1850845.
Gaildrat P, Killian A, Martins A, Tournier I, Frébourg T, Tosi M (2010) Use of splicing reporter minigene assay to evaluate the effect on splicing of unclassified genetic variants. Methods Mol Biol 653:249-257, PMID: 20721748.
With no in-frame deletion and likely NMD, the authors are left with two new null MPDZ alleles – and the task of explaining why the phenotypes in their proband are significantly milder than those of previously reported cases with biallelic null alleles. As written, their three explanations (lines 563-571) make no sense.
Quoting the ClinGen variant curation process SOP v2.0 (Jan 2021) document,
Many VCEPs (variant curation expert panels) have specified the splice predictor tools that are to be used in variant classification for the gene in question, as described below.
● MaxEntScan (http://genes.mit.edu/burgelab/maxent/Xmaxentscan_scoreseq.html) is based on the ‘Maximum Entropy Principle’ and utilizes probabilistic models of short sequence motifs to account for non-adjacent and adjacent dependencies between nucleotide positions. The score generated is the difference between a reference allele and a variant, and a higher score implies a higher probability of a true splice site.
● NNSPLICE (http://www.fruitfly.org/seq_tools/splice.html) is part of the Berkeley Drosophila Genome Project and utilizes a neural network method trained to recognize 5’ and 3’ eukaryotic splice sites using a representative data set from D. melanogaster. The score ranges from 0-1, with anything above 0.5 indicative of a possible splice site gain.
● SpliceAI6 is a deep neural network based on pre-mRNA transcript sequences that predicts splice sites using long-range primary genomic sequence flanking each position a input (+/-50 bp as default; +/-10,000 bp maximum). The user inputs HGVS nomenclature for the variant (https://spliceailookup.broadinstitute.org/). SpliceAI provides a table with delta scores (0-1) for acceptor loss, donor loss, acceptor gain, and donor gain within the designated flanking sequence. The delta score indicates the probability that the variant will alter splicing at the pre-mRNA position indicated.
● VarSEAK’s JSI splice site prediction tool (https://varseak.bio/) predicts splicing effects for genetic variants based on canonical splice site sequences (core motif GT for 5' donor splice sites or AG for 3' acceptor splice sites). The user enters the gene name, transcript, and variant. Output includes a graphical representation of the normal and variant sequence with annotated splicing impact, the overall splicing prediction class (1 for no splicing effect, to 5 for splicing effect) and a table with relevant splicing positions including the splice site prediction score, and ENT and delta ENT scores from MaxEntScan.
3. The zebrafish experiments involving truncated MPDZ forms were irrelevant and removed. However, the loss-of-function crispant data are nominally relevant to the paper, so were retained (Figs 4-5) What is “WT gRNAs+Cas9”?
Note correct “crispant” spelling (see https://www.nature.com/articles/s41684-021-00739-6).
4. The paper still needs extensive, detailed editing for English grammar, spelling, and style.
5. The PRDM13 primers and details of the PCR exclusion analysis should be included.
Round 3
Reviewer 1 Report
The most recent edits bring the paper up to an acceptable level for publication. However,
1. It still requires much editing for style and English grammar -- despite the attestation in the MDPI "certificate".
2. There is still no credible explanation for the comparatively mild phenotype, for having two null alleles in this important gene. There is some mention of unlinked modifier genes, which might (in principle) compensate for the brain ventricular developmental defects (hydrocephalus) predicted in the proband. However, this hypothesis is not seriously discussed.